# The Time Sequence of Gene Expression Changes after Spinal Cord Injury

**DOI:** 10.3390/cells11142236

**Published:** 2022-07-18

**Authors:** Seyoung Mun, Kyudong Han, Jung Keun Hyun

**Affiliations:** 1Department of Nanobiomedical Science & BK21 NBM Global Research Center for Regenerative Medicine, Dankook University, Cheonan 31116, Korea; 12200281@dankook.ac.kr; 2Center for Bio Medical Engineering Core Facility, Dankook University, Cheonan 31116, Korea; jim97@dankook.ac.kr; 3Department of Microbiology, College of Science & Technology, Dankook University, Cheonan 31116, Korea; 4Department of Rehabilitation Medicine, College of Medicine, Dankook University, Cheonan 31116, Korea; 5Institute of Tissue Regeneration Engineering (ITREN), Dankook University, Cheonan 31116, Korea

**Keywords:** spinal cord injury, RNA sequencing, time sequence, small molecules, gene ontology

## Abstract

Gene expression changes following spinal cord injury (SCI) are time-dependent, and an accurate understanding of these changes can be crucial in determining time-based treatment options in a clinical setting. We performed RNA sequencing of the contused spinal cord of rats at five different time points from the very acute to chronic stages (1 hour, 1 day, 1 week, 1 month, and 3 months) following SCI. We identified differentially expressed genes (DEGs) and Gene Ontology (GO) terms at each time point, and 14,257 genes were commonly expressed at all time points. The biological process of the inflammatory response was increased at 1 hour and 1 day, and the cellular component of the integral component of the synaptic membrane was increased at 1 day. DEGs associated with cell activation and the innate immune response were highly enriched at 1 week and 1 month, respectively. A total of 2841 DEGs were differentially expressed at any of the five time points, and 18 genes (17 upregulated and 1 downregulated) showed common expression differences at all time points. We found that interleukin signaling, neutrophil degranulation, eukaryotic translation, collagen degradation, LGI–ADAM interactions, GABA receptor, and L1CAM-ankyrin interactions were prominent after SCI depending on the time post injury. We also performed gene–drug network analysis and found several potential antagonists and agonists which can be used to treat SCI. We expect to discover effective treatments in the clinical field through further studies revealing the efficacy and safety of potential drugs.

## 1. Introduction

Spinal cord injury (SCI) is one of the most devastating lesions and is difficult to treat in a clinical setting. Many previous experimental studies using stem cells and biomaterials have shown some promising results [1,2,3]; however, the clinical translation of these treatments has not been elucidated due to safety and efficacy issues and insufficient evidence [4,5]. The pharmacological approach for spinal cord repair has been established since the 1970s with corticosteroids, but their clinical safety and effectiveness are still controversial [6]. Other drugs, such as minocycline, riluzole, and cethrin, did not show any promising evidence for clinical use [7].

Analysis of injury-induced genetic changes might enable us to detect various changes in factors in the microenvironment. RNA sequencing is superior to microarray in the detection of the transcriptome, offers an increased dynamic range [8], and might improve the prediction of diseases for clinical purposes [9]. Genes related to functional restoration following SCI are classified into several categories; for example, genes related to spinal cord development and regeneration, neuronal plasticity, inflammatory responses, macrophages, and glial scar formation, including proteoglycans. Because these genes are expressed variably according to the time post injury, it may be very important to verify factor changes based on the pathophysiology, which varies with time, and this is the first step to design optimal treatments for SCI. Previous studies found changes in some featured genes at the acute and subacute stages in spinal cord contusion models through RNA sequencing in mice [10] and rats [11,12]. Li et al. performed RNA sequencing at the acute stage (1, 4, and 7 days), and reported that some SCI-related genes including C5ar1, Socs3, CCL6, and Itgb2 are upregulated [11]. Shi et al. described various biological processes and pathways including immune response and chemokine signaling pathway based on RNA sequencing data which were obtained from spinal cord tissues at three different time points (1 day, 6 days, and 28 days) after SCI [12]. However, these studies have only found gene expression changes through RNA sequencing and have not suggested any drugs or genetic change-related treatments that can be used in a clinical setting.

We aimed to detect systematic gene expression changes after SCI within a time sequence by dividing it into five time points (1 hour, 1 day, 1 week, 1 month, and 3 months) from the very acute to chronic stages. Using RNA sequencing, we identify key genes and pathways and suggest potential drugs that might be applied to SCI treatment in a clinical setting.

## 2. Materials and Methods

### 2.1. Animals

Adult female Sprague–Dawley rats (12 weeks old, 230–250 g) were housed individually in a specific pathogen-free room with a 12:12 h light-dark cycle and free access to food and water. The animals were divided into 2 groups: the sham group (*n* = 45) and the SCI group (*n* = 45). Each group was divided into 5 subgroups according to the time sequence following SCI: 1 hour (*n* = 9), 1 day (*n* = 9), 1 week (*n* = 9), 1 month (*n* = 9), and 3 months (*n* = 9). Functional analyses were performed to all animals. Six rats from each group were randomly selected, and three rats were used for RNA seq (*n* = 3) and other 3 rats were used for histological analyses (*n* = 3).

### 2.2. Surgical Procedures and Tissue Preparation

All animal protocols were approved by the Institutional Animal Care and Use Committee of Dankook University (Approval No. DKU-13-016). Rats were initially anesthetized with 5% isoflurane (Forane; Choongwae Pharma, Seoul, Korea) inhalation and maintained with 1.5–2% isoflurane in 70% nitrous oxide (N_2_O) and 30% oxygen (O_2_). Laminectomy was performed at level T9-10 to expose the spinal cord and then the rats were subjected to a spinal cord contusion injury (200 kdyn force, 1200 μm displacement) using the Infinite Horizons impactor (IH impactor; Precision Systems and Instrumentation, Lexington, KY, USA). To verify reproducibility between animals, animals showing a value more or less than 5% based on the actual force measured at 200 kd and displacement at 1200 μm were excluded. Sham-operated rats received only laminectomy without spinal cord contusion injury as previously described [13]. After surgery, all operated rats were injected intraperitoneally with normal saline (3 mL) and received intramuscular injection of 40 mg/kg cefotiam hydrochloride (Fontiam™; Hanmi Pharma, Seoul, Korea) and oral administration of 10 mg/kg acetaminophen syrup (Tylenol™; Janssen Pharmaceutical, Titusville, NJ, USA) once a day for 3 days. Manual bladder expression of SCI rats was performed twice a day until the amount of expressed urine was less than 0.5 mL/day. For spinal cord tissue preparation, all animals were deeply anesthetized with an overdose of 50 mg/kg ketamine hydrochloride (Ketamine; Yuhan, Seoul, Korea) and 25 mg/kg xylazine hydrochloride (Rumpun; Bayer Korea Ltd., Seoul, Korea) and perfused with ice-cold DNase, RNase, and protease-free PBS (Thermo Fisher Scientific Inc., Waltham, MA, USA), and spinal cords were dissected (5 mm rostral and 5 mm caudal to the lesion site, 10 mm long) and immediately frozen in liquid nitrogen (LN2) until RNA isolation.

### 2.3. Histology

For histological analysis, all animals were deeply anesthetized and transcardially perfused with 150 mL of saline, followed by 500 mL of 4% paraformaldehyde (PFA) in 0.1 M PBS (pH 7.2) via a peristaltic pump. After perfusion, the spinal cord was removed, postfixed in 4% PFA overnight, and cryoprotected in 30% sucrose for 3 days. The tissues were embedded in M1 compound (Thermo Fisher Scientific Inc., Waltham, MA, USA) and sectioned sagittally at a thickness of 16 µm on a cryostat. H&E staining was performed at each time point after SCI. Sections were washed in 0.1 M PBS and immersed in Harris hematoxylin solution (Merck KGaA, Darmstadt, Germany) for 3 min followed by a brief wash in distilled water. Slides were then immersed briefly in 1% acid alcohol (1% HCl in 70% ethanol) and stained with eosin Y solution (BBC biochemical, Mount Vernon, WA, USA) for 30 s. The slides were dehydrated with an ethanol series, cleared with xylene, mounted in DPX (Merck KGaA, Darmstadt, Germany), and observed under a microscope (Nikon, Tokyo, Japan).

### 2.4. Locomotor Assessment

Locomotor function of the hindlimbs was scored using the Basso, Beattie, and Bresnahan (BBB) locomotor rating scale and horizontal ladder test by two blinded observers who were positioned across from each other to observe both sides of the rats during a 4-min walk in a cylindrical-shaped acrylic box (90 cm diameter, 15 cm high with a smooth floor, BBB scale) or an acrylic wall-installed runway (10 cm tall, 127 cm long, 8 cm wide between walls, 1 cm between rungs, horizontal ladder test). The calculation of BBB and ladder scores was performed as previously described [13,14]. For horizontal ladder test, all animals were trained to walk from left to right on a runway several times for adaptation before testing and then test was performed on a runway made of acryl walls. The horizontal ladder score was calculated as the number of erroneous steps of hindlimb divided by total number of steps of hindlimb (%). Locomotor function was examined every 7 days until sacrifice. In this experiment, animals that obtained a BBB score of 2 or more points at 2 days after injury were considered to have insufficient initial injury and were excluded.

### 2.5. RNA Sample Preparation

The time-dependent tissue samples were separated and homogenized in 500 µL of TRIzol reagent (Invitrogen, Carlsbad, CA, USA) using a microhomogenizer following the manufacturer’s instructions. Total RNA was isolated from frozen spinal cord tissue using the RNeasy Mini Kit (Qiagen, Hilden, Germany).

### 2.6. RNA Sequencing Library Construction

Prior to constructing RNA sequencing libraries, the quality of all RNA samples was checked using the 28S/18S ratio and RNA integrity number (RIN) value using an Agilent Bioanalyzer 2100 system (Agilent Technologies, Santa Clara, CA, USA). All RNA samples obtained from each stage and case showed RIN values higher than 8.0. mRNA molecules were enriched and purified from 500 ng of the qualified RNA samples using oligo-dT magnetic beads. Double-stranded cDNA was immediately synthesized by SuperScript III reverse transcriptase (Thermo Fisher Scientific Inc., Waltham, MA, USA). According to the instructions of the TruSeq RNA Sample Prep Kit (Illumina, San Diego, CA, USA), a sequential process of end repair, poly-A addition, and adaptor ligation on both ends was carried out. The processed cDNA libraries were subjected to library enrichment by polymerase chain reaction (PCR) and size selection to the exact appropriate size of fragments using the BluePippin Size-Selection system (Sage Science, Beverly, MA, USA). The final selected libraries were evaluated with an Agilent Bioanalyzer 2100 system and were 400–500 bp in size. The cDNA libraries were sequenced with an Illumina HiSeq2500 (IlluminaSan Diego, CA, USA), which generated paired-end reads of approximately 100 bp in size.

### 2.7. Data Analysis

#### 2.7.1. Quality Control

Raw sequencing data were evaluated to discard low-quality reads by FAST-QC (https://www.bioinformatics.babraham.ac.uk/ accessed on 8 November 2020) as follows: reads including more than 10% skipped bases (marked as ‘N’s), sequencing reads including more than 40% bases whose quality score was less than 20, and average quality score <20. Quality distributions of nucleotides, GC contents, the proportions of PCR duplication, and k-mer frequencies of sequencing data were also calculated [15].

#### 2.7.2. Read Mapping and Differentially Expressed Genes (DEG) Analysis

To increase mapping quality, only highly qualified reads were mapped to the rat reference genome (Rattus norvegicus: Rnor 5.0) using the aligner STAR v2.4.0 [16]. We only used uniquely mapped read pairs for the analysis of downstream DEGs. To identify DEGs, gene expression count data were generated using HTSeq-count v0.6 [17], and the TCC R package for comparing fragments per kilobase of exon per million fragments (FPKM) count with the normalization method [18] was used. Differentially expressed genes (DEGs) at the 5 time points were analyzed using DESeq2 v1.26.0 [19]. DEGs with log2-fold-change (FC) greater than 1 and adjusted *p*-value (FDR) less than 0.05 were considered statistically significant.

#### 2.7.3. Statistical Analysis and Visualization of Data

For the expression data across all samples, the log2 transformed FPKM values were represented by qualitative characteristics of normalized data, including the count distributions and variability between biological replicates. The general analysis for statistical validation, including pairwise correlation analysis and scatterplot, hierarchical clustering heatmaps, and principal components analysis (PCA) plots were performed with the ggplot2 package. The heatmap clustering analysis of DEGs was performed based on the log2 FPKM values, and the heatmap was generated using the hclust2 package (v 3.6.2, available at https://github.com/SegataLab/hclust2 accessed on 10 January 2021) with the popular clustering distance (Euclidean) and hierarchical clustering method (complete) functions. A Venn diagram was generated using jVenn (http://jvenn.toulouse.inra.fr/app/index.html accessed on 20 January 2021) [20].

#### 2.7.4. Integrative Function Classification Analysis for DEGs

Gene ontology (GO) and enrichment analyses of time-dependent DEGs were performed using Metascape (http://metascape.org/gp/index.html accessed on 6 February 2021). The Metascape analysis workflow followed these criteria. First, the multiple gene lists identified from DEG analysis were used as input genes. Second, for GO annotation, three main categories of gene functions (biological process (BP), molecular function (MF), and cellular component (CC)) were extracted. Third, functional enrichment analysis was performed with default parameters (min overlap of 3, enrichment factor of 1.5, and *p*-value of 0.01) for filtering [21]. The Reactome pathway database, which interprets biological pathways, was also used to identify the functional role of genes that show differences in expression depending on injury periods. The protein–protein interaction (PPI) network was constructed and visualized using STRING software.

#### 2.7.5. Time-Series DPGP Clustering Analysis

We performed time-course gene expression clustering analysis using the Dirichlet process Gaussian process mixture model using the DP_GP_Cluster Python package to cluster the DEGs that showed a significant expression difference (same with DEG cut off) between sequential time points [22]. The required Python packages (V 2.7.12): Cython (V0.29.23), GPy (V 0.8.8), Matplotlib (V 2.2.5), numpy (V 1.16.6), pandas (V 0.24.2), paramz (V 0.9.5), scikit-learn (V 0.20.4), and scipy (V 1.2.3) were installed and set. The significant log2 FC value satisfying the statistically adjusted *p*-value cutoff was only used for this analysis as the input data. The optional arguments follow: -n 100 --plot –fast was used to cluster genes by expression over the time course and create a gene-by-gene posterior similarity matrix. After constructing the posterior similarity matrix, the plotting options were as follows: --clusterings, --criterion MPEAR, and --post_process –plot were used for creating the gene expression trajectories and a heatmap with dendrogram by cluster. The cluster number was set to 22, maximizing the total number of subcategories of DEGs enriched for the transcriptome data.

#### 2.7.6. Gene–Drug Network Analysis

All DEGs, including key drivers, were used to find potential drugs. We used the drug repurposing hub (DHUB) dataset that could be obtained from https://clue.io/repurposing accessed on 20 October 2021. The DHUB dataset was suitable for substituting the findings of this study because it provides information on comprehensive FDA-approved drugs, clinical trial drugs, and preclinical tool compounds for all nonspecific diseases, such as cancer [23]. The dataset introduces information on 6798 drugs or compounds with 2183 genetic targets covering 24 disease areas. In addition, by providing the mechanism of action (MOA) for drugs, the dataset made it possible to distinguish compounds according to antagonist and agonist efficacy dependent on DEGs after SCI. In more detail, as shown in Appendix A, we took into account (i) genes; (ii) drugs; (iii) clinical phase, and (iv) disease area and their indications. Here, we used Cytoscape software (version: 3.9.1) to construct the drug–gene interaction network [24]. Only the drugs with ample evidence supporting significant interactions with genes were involved in the results of the drug–gene network analysis.

## 3. Results

### 3.1. Histological and Functional Results of SCI Models at Five Time Points

Histological findings showed that a lesion cavity was formed from 1 day after contusion injury and continued until 3 months (Figure 1A). The BBB score was 6.12 ± 1.79 at 1 week and increased to 10.79 ± 0.73 at 1 month, and the final score was 11.68 ± 0.94 at 3 months (Figure 1B). The ladder score started to recover from 2 weeks after contusion injury (93.21 ± 4.98) and reached a plateau at 6 weeks (68.47 ± 15.69), and the final score was 66.20 ± 15.71 at 3 months (Figure 1C).

### 3.2. Transcriptome Sequencing Analysis

To determine the transcriptional change between the SCI contusion and sham groups at the five time points, a total of 30 spinal cord tissue samples (three samples per group and time point) were subjected to RNA sequencing analysis. Using an Illumina HiSeq 2500, an average of 46.4 million raw reads were produced, with a read length of 100 bp. Approximately 94.6% of the raw data were qualified through the quality control step for sequence data, and the average number of 39.6 million reads, accounting for 85.4% of raw data, was uniquely mapped on the rat genome (Rnor 5.0) (Appendix A). After gene annotation using the Ensembl database (release 77) (http://oct2014.archive.ensembl.org/index.html accessed on 15 December 2020), among the 26,689 reference genes, a total of 20,187 were detected in at least one group, and 14,257 were commonly expressed at all time points (Appendix A). The distribution of expressed genes in ten cases showed a consistent expression level, supporting that there was no variability in sample preparation or data generation. We confirmed the uniform data conditions in the gene expression distribution for each sample. (Appendix A). To emphasize the association between samples in each group, the reproducibility of technical replication using scatterplot and pairwise correlation analysis was confirmed based on the overall gene expression in the sample. As a result, similar distributions within the group appeared solid, and distinct differences between groups were confirmed (Appendix A). For the sham group, the gene expression pattern of all sham samples at the different injury times were identical, while that in the contusion group was clearly divided into five time points (1 hour, 1 day, 1 week, 1 month, and 3 months).

The log_2_ transformed FPKM was calculated to determine the similarity between samples. A principal component analysis (PCA) plot can describe the degree of variability of the entire dataset. The results showed that our preclinical testing of injury treatment strategies (contusion injury and sham surgery for the spinal cord) clearly separated each group. The reproducibility and concordant transcriptome alterations between replicates were demonstrated (Appendix A). Moreover, the hierarchical clustering heatmap also showed transcriptional concordance among the replicates (Appendix A).

### 3.3. Transcriptional Waves According to the Time after SCI

#### 3.3.1. Identification of DEGs in Each Period and Their Functional Prediction

Our comprehensive analysis of DEGs revealed that changes in transcriptional waves over the five time points after SCI occurred demonstrate the characteristics of critical biological processes. To identify DEGs in the spinal cords of contusion- and sham-injured rats, gene expression data from each group were compared using DEGSeq2 software. DEGs were defined by strict cutoff criteria as follows: at least a 2-fold change (log_2_ FC ≥ 1) in relative transcription levels with an adjusted *p*-value and *q*-value < 0.05.

As a result of comprehensively analyzing the Metascape GO function prediction analysis for changes in gene expression during all periods from 1 hour to 3 months, we were able to assume that there is a flow of biological changes depending on each timepoint, as shown in Figure 2A and Figure 3B. Furthermore, the upregulated and downregulated genes with expression changes at all time points were biased toward the gene sets ubiquitous in the inflammatory and immune systems and the nervous system, respectively (Figure 2C). In the comparative analysis of spinal cord tissue at 1 hour with contusion vs. sham groups, we identified 187 DEGs (181 upregulated and six downregulated) after SCI. Brief functional classification of DEGs at 1 hour revealed that 151 out of 187 DEGs were primarily associated with the inflammatory response (GO:0006954). In particular, the difference in expression of the top 10 genes, including *Il6*, *Jun*, *Tnf*, *Il1b*, *Egr1*, *Anxa1*, *Serpine1*, *Myc*, *Ccl2*, and *Il1a*, showed the highest relevance to the immediate biological response at 1 hour after SCI. One day after SCI, a total of 795 (472 upregulated and 323 downregulated) DEGs showed significant expression changes. Three hundred sixty-three genes were highly enriched in the biological process (BP) of inflammatory response and cellular component (CC) of integral component of synaptic membrane (GO:0099699). The up- and downregulated genes were separately involved in the inflammatory and synaptic biological pathways. The top 10 function-related genes were listed as *Il1b*, *Il6*, *Ccl2*, *Nos3*, *Serpine1*, *Hmox1*, *Anxa1*, *F3*, *Cd36*, and *Mmp9*. One week after SCI, 750 out of 1558 (1123 up- and 435 downregulated) DEGs were highly enriched in the BP of cell activation (GO:0001775) and the CC of presynapse (GO:0098793). The top 10 genes were shown to be more strongly associated with those biological activities included *Syk*, *Anxa1*, *Lyn*, *Fcer1 g*, *Tgfb1*, *Pycard*, *F2rl1*, *Cd40*, *Myd88*, and *Trem2*. One month after SCI, 750 out of 1509 (947 upregulated and 562 downregulated) DEGs were highly enriched in the CC of presynapse and cytosolic ribosome (GO: 0022626) and the function of the innate immune response (GO: 0045087). The top 10 function-related genes were listed as *Fcer1* g, *Lyn*, *Anxa1*, *Syk*, *Tlr2*, *Pycard*, *Tgfb1*, *Unc13d*, *Btk*, and *Fcgr2a*. Three months after SCI, 581 out of 1088 (888 upregulated and 200 downregulated) DEGs were shown to be highly enriched in the BP of cell activation and regulation of cell activation (GO: 0050865). The top 10 function-related genes were listed as *Lyn*, *Fcer1 g*, *Syk*, *Anxa1*, *Pycard*, *Trem2*, *Lbp*, *Tgfb1*, *Fcgr2a*, and *Btk* (Figure 2A,B, and Appendix A). In the comparison analysis of every time point in contusion vs. sham-treated spinal cord samples, a total of 2841 gene sets differentially expressed at any of the five timepoints were identified, as listed in Appendix A. The number of DEGs across every injury period was visualized using a Venn diagram, and the degrees of expression differences were presented using statistical plots, as shown in Appendix A.

#### 3.3.2. Systematic DEG Classification Using the DPGP Clustering Method for Time Series Analysis

To reconfirm and characterize the time-dependent diversity of the transcriptional response to SCI, we next used the Dirichlet process Gaussian process mixture model (DPGP) clustering analysis, which is able to recover actual cluster structure across a variety of generating assumptions except in cases of a large number of clusters, each with a small number of genes [22]. DPGP clustering analysis with the given 2841 DEGs resulted in 22 clusters with a mean size of 129 DEGs (minimum genes per cluster = 20; maximum = 768). Of 22 clusters, cluster 5 had the most extensive dynamic trajectory of expression change, with 768 DEGs during the subacute and chronic phases. To analyze the specific mechanisms underlying the expression dynamics for genes within a cluster and to validate cluster membership, we chose nine modules representing the up- and down-trajectory at each time point from 17 out of 22 clusters (Appendix A). Five modules with 10 clusters having a specific upward trajectory in each period were divided into 871 DEGs (1 hour: 72, 1 day: 219, 1 week: 398, 1 month: 133, and 3 months: 49) (Figure 3A,C). The other four modules with seven clusters having a downward trajectory in each period were divided into 553 DEGs (1 hour: 47, 1 day: 146, 1 week: 303, and 1 month: 57) (Figure 3B,D). We found that any cluster with a downregulation pattern at 3 months could not be represented because most of the SCI-induced downregulation of DEGs started less than 3 months earlier (from 1 hour to 1 month). In addition to the 17 clusters, we reveal long-term fixed expression changes from the very acute to chronic stages and a fluctuating multifold expression change across the five time points. In Cluster 5, 758 DEGs were fixed in high expression over a long period from 1 hour to 3 months. A total of 127 and 191 DEGs underwent a static decrease in transcriptional expression from 1 day to 3 months and from 1 week to 3 months, respectively (Appendix A).

In addition, 272 DEGs showed the highest trajectory for one week and showed a high difference in expression, but an interesting pattern was confirmed with a rapidly decreasing pattern immediately after one month (Figure 4). We reconfirmed the biological functions of the genes involved in the identified time-series diversity using DPGP clustering with the GO function prediction results of independent DEGs at each time period. Modules in which nine specific up- and downregulated expression changes were observed for each period were identified. Despite the transcriptional upregulation of genes specific to each period, most of the functions involved in biological pathways that resisted external stimuli, such as inflammatory response (GO:0006954) and positive regulation of cell death (GO:0010942), were highest. Forty-three genes related to cytoplasmic translation (GO:0002181) displayed high enrichment scores and were involved in the significant transcriptional changes 1 month after SCI. Modules with downregulation over the five time points confirmed that key genetic factors, such as chemical synaptic transmission (GO:0002181) and ion transmission regulation (GO:0043269), were staged in the development of the nervous system and neurotransmission (Figure 4). We designated the modules ‘Acute-Chro_UP’ (1 hour to 3 months), ‘Acute-Chro_DN_1’ (1 hour to 3 months), ‘Acute-Chro_DN_2’ (1 day to 3 months), and ‘Fluctuating’ (no defined pattern) for the extra module representing the long term. As a result of specific upregulation over the five time periods, functions such as the inflammatory response (GO:0006954) and regulation of defense response (GO:0031347) accumulated for the Acute-Chro_UP module (Appendix A). Acute-Chro_DN_1 and Acute-Chro_DN_2 were also enriched for terms related to the regulation of ion transport (GO:0043269) and chemical synaptic transmission (GO:0007268) (Appendix A). It was interesting to note that the fluctuating modules, such as the modules showing long-term downregulation, were involved in similar biological functions and confirmed more significant associations (Appendix A). Typically, *ASAH2*, *DERS4*, and *SGPP2* are involved in sphingolipid metabolism, which is abundant in the central nervous system (CNS), participates in tissue development, cell recognition, and adhesion, acts as a receptor for toxins, and is included in the trajectory of the fluctuating module [25]. Together with the results of our DPGP clustering analysis, we provide fascinating insights into the gradual or cascading changes derived from contused spinal cords from 1 hour to 3 months. The detailed information for each cluster module is categorized in Appendix A.

### 3.4. Molecular Changes Caused by SCI

We conducted an integrated analysis to determine the molecular differences that occur immediately or continuously according to phase in spinal cord tissue after SCI. In each period, all up- and downregulated genes were contrasted using Venn diagrams (Figure 5A,B). As a result, there was a significant difference in the number of DEGs in each period, and as a representative example, the most specific expression difference could be confirmed in the 1-week spinal cord sample. This result indirectly suggested that transcriptional alterations in the 1-week sample might be a precious stage allowing for clinical improvements such as axonal regrowth, neurogenesis, and functional recovery. We were able to confirm that many genes were shared between both stages every time the stage passed. Moreover, as a result of ascertaining the correlation between samples through pairwise distance calculation, it was confirmed that the total of 2841 DEGs identified in our study were faithfully divided into the very acute (1 hour), acute (1 day), and subacute to chronic stages (1 week to 3 months) (Figure 5C). Interestingly, with the same process, the sham groups were found to have identical aspects of gene expression across all stages, suggesting that our strategy of experimentation with a local spinal injury and complete-crush transection of the spinal cord is very reliable for verifying spinal cord tissue-specific differences. The hierarchical clustering heatmap results showed high similarities among the replicates of independent period conditions (Figure 5D).

Among the 2841 DEGs, 18 genes (17 upregulated and one downregulated) showed common expression differences at all time points (Figure 5A,B). Interestingly, we found that 12 of the 17 upregulated genes were closely intersected by protein–protein interactions (PPIs), and the seven genes, including activating transcription factor-3 (*ATF-3*), containing basic-leucine zipper (bZIP) transcription factor domains (Figure 6A) [26]. *ATF3* is a transcription factor called the immediate early gene, inductive stress gene, and adaptive response gene, known as a protein molecule that associates biological reactions with injury damage and cellular stress [27]. Coexpression of *ATF3* with *Jun/AP-1*, *ATF2*, *ATF4*, *ATF6*, *CREB*, *Myc*, *C/EBPB*, and *Erg-1* suppresses oxidative stress-induced apoptotic cell death during the early stage of the response, thereby inhibiting proinflammatory cytokine expression and is a major mechanism for controlling inflammatory reactions by controlling anti-inflammatory cytokines [27]. Our results suggest that *ATF3*, *JUNB*, *MAFF*, *CEBPD*, and *MYC*, which play significant roles in regulating cytokines that cause innate inflammatory responses arising from SCI, showed higher expression than the sham control during all experimental periods, especially in the acute phase (1 hour and 1 day) (Figure 6A). In addition, it was also confirmed that *JUN* and *JUND* showed very high expression in the acute phase, although they did not show significant expression at all time periods. Together with *ATF3*, upregulation of *JUNB*, *MAFF*, *CEBPD*, and *MYC* could have pro-regenerative neuronal effects in SCI. Importantly, we first reported that *SLC17A7* (VGLUT1), which is known to play a pivotal role in absorbing glutamate into synaptic vesicles at presynaptic nerve terminals and excitatory neurons, was identified as one of the molecules with significant deficiencies in SCI as evidenced by downregulation in all SCI contusion time points (Figure 6B). Glutamate is an essential central neurotransmitter in the amino acid family, along with GABA and glycine. Thus, its promotion of *SLC17A7* expression might be a pivotal point in neuronal damage control, recovery, and protection [28].

### 3.5. Exploration of the Molecular Biological Cascade through Integrated Analysis of the Continuous Period after SCI

To more specifically explore the biological pathway and reactions occurring in each time period, we integrated and analyzed genes with expression differences specific to every period using the Reactome database version 77 [29]. A total of 1698 out of 2493 identifiers in the sample were found in Reactome, where 2120 pathways were hit by at least one of them. Among them, the top 20 Reactome pathways were significantly enriched with a higher number of gene entities (*p*-value < 0.005) (Appendix A). We analyzed cascades of biological changes caused by SCI, focusing on eight remarkable functional pathways, including the ‘translation termination’ pathway, associated with 13 subgroups. A total of 358 genes were enriched in these pathways. The most significant pathway was ‘neutrophil degranulation,’ with 166 genes (*p*-value = 4.76 × 10^−13^). Other significant pathways were ‘eukaryotic translation termination,’ ‘LGI–ADAM interaction,’ ‘Interaction between L1 and Ankyrina,’ ‘Interleukin-4 and Interleukin-13 signaling,’ ‘Interleukin-10 signaling,’ ‘GABA receptor activation,’ and ‘Collagen degradation.’

As shown in Figure 7, all DEGs that show a significant relationship to the Reactome pathway are at the center of dramatic change at each stage of SCI, and their changes are reduced as the chronic stage continues. Volcano plots explained the transcriptional eruption of significant biological pathways through Reactome analysis. Through this result, it can be confirmed how significantly the differences in expression of biological pathways at the time of SCI were shown. Notably, the most significant changes were observed at 1 week (Figure 7B) and 1 month (Figure 7D). After examining the changes in multilateral transcriptional expression through pathway analysis, a very immediate immune response occurred after SCI, and the changes in related key factors including IL-1A, IL-1B, CCL4, FOS, and TNF were clearly identified as described following subchapters (Section 3.5.1, Section 3.5.2 and Section 3.5.3).

Neutrophils and macrophages play a central role in the inflammatory response to infection and tissue damage, and various acute spinal cord tissue injury stimulants can trigger molecular biological pathways involved in neutrophil and macrophage activation [30]. Especially in the acute phase, the 166 genes involved in the ‘neutrophil degranulation’ pathway began to show conspicuous changes in gene expression iteratively during SCI progression from acute to chronic, consistent with previous tissue injury-induced associations with neutrophil and macrophage activity. Starting with the explosive increase in genes related to hemoglobin (*HBA2*, *HBA1*, *HBB*, and *HBB-B1*) and microglial cell immune response (*CXCL1*, *OLR1*, *TLR2*, and *GPR84*) 1 hour after SCI, we observed that the increase and degranulation of neutrophils in acute inflammation resulting from tissue damage were immediately presented to remove pathogens from the damaged lesion and control the inflammatory response (Appendix A).

#### 3.5.1. Interleukin Signaling

The ‘Interleukin-4 and Interleukin-13 signaling’ and ‘Interleukin-10 signaling’ -related pathways were also enriched with the upregulation of 65 particular genes, which play a significant role in macrophage polarization. The 65 major regulators of inflammation provided evidence that they had a competitive relationship in the formation and regulation of M1/M2 macrophages in molecular biological changes after SCI. We found that M1 macrophage activation was induced via upregulation of immunostimulant factors such as proinflammatory cytokines (*IL-1*, *IL-6*, *IL-18*, and *TNF*) and chemokines (*CCL2*, *CCL3*, and *CCL4*) initially at 1 hour and 1 day after SCI. Among the interleukin signaling-related genes, IL-1A, IL-1B, CCL4, FOS, and TNF presented the highest expression differences at 1 day after SCI. Next, anti-inflammatory cytokines such as IL-4, IL-10, IL-13, and TGF-b1, related to the formation of M2 macrophages, showed a different pattern at 1 week. The inflammatory response continues to increase until 3 months due to the overexpression of proinflammatory cytokines, but the polarization of M2 macrophages is also ongoing as a mechanism to mediate this competitivity. *IL-10RA*, *IL-10RB*, *TGF-B1*, *STAT6*, and *TIMP-1,* as representative anti-inflammatory molecules, increased rapidly 1 week after SCI, which is generally thought to alleviate tissue and cell damage by suppressing inflammation and promoting beneficial matrix remodeling and repair through the release of regeneration and anti-inflammatory factors and the formation of M2 macrophages [31].

#### 3.5.2. Neutrophil Degranulation, Eukaryotic Translation, and Collagen Degradation

In addition to the pathways above, we also derived the results of ‘Eukaryotic translation termination’ and ‘Collagen degradation’ for the subpathways that might be affected due to an excessive immune response (Figure 4 and Appendix A). The occurrence of reactive oxygen species (ROS) by neutrophil degranulation in the human body is important for pathogen killing and serves as a signaling molecule and mediator for inflammation, while fragmentary excess ROS and deficient antioxidants result in inflammatory tissue damage [32]. The cascading biological disruption of ROS resulting from excessive neutrophil degranulation could lead to negative effects, such as translation alteration and promotion of collagen degradation [33,34]. We observed high expression of 58 ribosomal protein (RP) gene families, including 23 small (RPS) and 35 large (RPL) subunits, which are essential elements for the translation process, for 1 month. Although injury stress stimulates the synthesis of abundant repair proteins to repair damaged tissues or cells with activation of translational processes, it is assumed that the increase in cell death or apoptosis caused by inflammation results in the silencing of ribosome synthesis in chronic inflammation [35]. In addition, interestingly, it was confirmed that collagen-degrading enzymes (Furin, Cathepsin, Matrix metalloproteinase (*MMP*) gene family) having a specific role in collagen degradation and fragmentation showed rapid upregulation after SCI. As a result, excessive immune response, neutrophil degranulation, ROS generation, etc., degrades the collagen component of the extracellular matrix and prevents the cells from surviving any longer, resulting in damage and apoptosis [33]. One day after SCI, the expression of *MMP3*, *MMP9*, and *MMP10* gradually increased. Decisive collagen degradation factors, including *CTSD*, *CTSB*, *CTSL*, *CTSK*, *MMP19*, and *MMP12*, remained upregulated from 1 week to 3 months. Based on these results, we suggest that proinflammatory cytokines in spinal cord injury contribute to chronic inflammatory conditions and adverse cell maintenance effects.

#### 3.5.3. LGI–ADAM Interactions, GABA Receptor, and L1CAM–Ankyrin Interactions

Our analysis revealed very particular biological pathways in nervous system development and function and synaptogenesis of cortical inhibitory (GABAergic) neurons, such as the ankyrin binding, mediate LGI–ADAM interactions (15 genes), GABA receptor (24 genes), and L1CAM- ankyrin interactions (10 genes) (Appendix A). First, a continuous decrease in *LGI1*, *LGI2*, *LGI3*, *ADAM11*, *ADAM22*, and *ADAM23* was observed 1 day after SCI. Synaptic formation and maturation require multiple interactions between presynaptic and postsynaptic neurons [36]. Among them, the LGI–ADAM interaction plays a vital role in the development and function of the spinal nervous system, mainly mediating synaptic transmission and myelination [37,38]. In this way, we have obtained evidence of deficiencies in the expression of LGI and ADAM, a class of molecules that are fundamental to cellular interactions in myelination and maintenance of the nervous system [39]. Although there is no in-depth study on the nerve regeneration and repair effects of LGI–ADAM interactions in the recovery from SCI, we believe that their absence will affect neuronal plasticity at the chronic stage in SCI from a molecular genetics point of view. Their interactions also suggested that members of the voltage-gated calcium channel y subunit gene family (*CACNG2*, *CACNG3*, and *CACNG8*), corresponding to an identical pattern in SCI, are also worth noting. Simultaneous with the lack of LGI–ADAM interactions, the expression of 8 out of 18 gamma-aminobutyric acid (GABA) receptor gene families closely related to ‘GABA receptor activation’ was rapidly decreased. This reflects the clear finding that postsynaptic GABA receptor activity is blocked downstream after SCI, disrupting LGI–ADAM interactions supporting synapse formation and maturation between pre- and postsynaptic neurons. The reduction in 9 heterotrimeric G-protein genes required to construct the GABA receptor complex, one of the G-protein coupled receptors (GPCRs), was also significantly confirmed. We identified that three adenylate cyclase (*ADCY*), a neural-specific protein that catalyzes cAMP production, and three potassium inwardly rectifying channel subfamily J (*KCNJ*) gene families, an ATP-sensitive potassium channel protein, were also downregulated in relation to GABA receptor activation. Among the GPCR genes essential for cell homeostasis, including ion channels, cell transcription, and neuronal secretion, 24 genes in the spinal cord showed a significant decrease in expression one week after SCI. In addition, L1CAM is an essential mediator of neurodifferentiation, including axonal growth, synaptic formation, and maintenance [40], and L1CAM and ankyrin (*ANK*) interactions play an essential role in neuroprotection, promoting axonal development and reducing outgrowth-inhibitory molecules near lesions [41]. However, the expression of the spectrin (*SPTA1*, *SPTB* and *SPTBN2*) and ankyrin (*ANK1*, *ANK2*, and *ANK3*) gene families, which are closely related to the mechanical support of the plasma membrane and the formation of membrane skeletons maintaining the stability and structure of the shape of a cell, such as synaptic microtubules, confirmed a sharp decrease with L1CAM 1 month after SCI. We traced back the molecular candidates, which might be essential for regulating membrane excitability and synaptic transmission in the normal state without SCI, and confirmed their abnormalities following SCI.

### 3.6. Gene–Drug Network Analysis Provides Potential Drug or Chemical Candidates for the Treatment of SCI

One of the purposes of this study was to focus on improving the knowledge of drug therapy potential by understanding the basic mechanisms of improvement after SCI, resulting in temporary or permanent cessation of somatic and autonomic dysfunctions below the neurological level of injury. To explore the potential therapeutic drugs related to or that may respond to transcriptional changes caused by SCI, we performed network analysis using the Drug repurposing hub (DHUB) dataset. A total of 2841 DEGs across five time points after SCI were implicated in the drug–gene network identification step using Python. Against the DHUB dataset, 3462 interactions between 498 genes and 1983 drugs were mined. The detailed data for raw gene-drug interactions are listed in Appendix A. These data include detailed information, such as preclinical and clinical trials and withdrawn drugs. Thus, we only focused on the currently launched drugs, whose efficacy and use have already been clearly identified, in the next step. As a result of entering the up- and downregulated genes in each period into the DHUB dataset and comparing it, 1374 interactions between 175 genes and 501 drugs and 1522 interactions between 146 genes and 410 drugs were confirmed. As the last step, considering the abnormally biased changes in gene expression caused by SCI, drug antagonists were selected from gene–drug interactions focused on upregulation, whereas agonists were selected focusing on downregulated DEGs. As a result, we derived 963 antagonistic interactions between 128 genes and 366 drugs and 384 agonistic interactions in 75 downregulated genes along with 116 drugs. The representative network with the highest connections across the five time points included dextromethorphan, procaine, amiloride, dasatinib, boceprevir, and bosutinib for upregulated DEGs. For downregulated DEGs, L-glutamic acid, dehydroepiandrosterone, isoflurane, propofol, and valproic acid were searched as potential agonist drugs that could prevent imbalances or synergize due to biological deficiency or reduction after SCI (Table 1). Evidence from prior research on the neuropharmacological, genetic, cytobiological, and histological perspectives of drugs administered to SCI models demonstrates how reliably our gene-pharmacological network analysis was determined. In the Gene–Drug network association, which we observed using the DHUB dataset, more results provide information on potential treatments or compounds that respond to complex biological changes at the tissue or cell level that occur after SCI (Appendix A). One hundred twenty antagonistic drugs were associated with 44 upregulated DEGs, and 93 agonistic drugs were associated with 58 downregulated DEGs (Figure 8 and Appendix A).

## 4. Discussion

In this study, we performed molecular analysis of the spinal cord at five time points: 1 hour, 1 day, 1 week, 1 month, and 3 months. These time points are not only pathophysiologically meaningful but also clinically significant periods. In the clinical field, initial treatment might be possible for SCI patients within 1 hour after injury according to the arrival time from the place of injury to the hospital by helicopter or emergency vehicles [42,43]. The timing of surgical intervention is also a critical issue for SCI. Early surgery, usually within 24 h, showed better neurological outcomes than late surgery in SCI patients [44,45] and shortened hospital stays [46], although some studies reported that early surgery without consideration of lesion severity and hemoglobin level may increase the mortality rate [47,48]. After acute management, mobilization and rehabilitation are begun in the intensive care unit within several days. Early rehabilitation is very important to lessen medical complications and the length of hospital stay [49], and in vivo studies have also revealed that early exercise enhances functional recovery after SCI [50]. The mean transfer time from injury to the active rehabilitation unit is a month, and the length of stay in the rehabilitation unit is between 33 and 74 days [51]. The recovery period after SCI can persist for up to 18 months, but major recovery occurs in the first 3 months [52]. Therefore, we can apply appropriate treatment strategies at each time point, which is critical clinically and biologically, through extensive biological and molecular analysis.

We were able to identify the time-series diversity of the transcriptional and biological responses to SCI through DPGP clustering analysis [22] and the Reactome database [29] as well as time-dependent analysis of gene expression changes. The upregulation of anti-inflammatory cytokines (IL-4, IL-10, and IL-13) after SCI is important for the polarization of M2 macrophages, and we also found that the enhancement of M2 macrophage polarization through the application of hydrogel containing decellularized porcine brain matrix promoted functional restoration after SCI in a previous study [53]. On the other hand, neutrophil degranulation, which might enhance ROS production and proinflammatory signals, leads to tissue damage after SCI, while increased numbers of neutrophils after SCI also act to neutralize pathogens [54]. The role of LGI and ADAM and their interactions after SCI have not yet been clarified; however, some previous studies have revealed their roles in the CNS. Xi et al. found that LGI1 promoted oligodendrocyte development and myelination in the brain [55], and Fukata et al. found that the LGI1-ADAM22 protein complex is important for the formation of synaptic transmission in the brain, and the loss of LGI1 and ADAM22 led to epilepsy [36]. Therefore, downregulation of LGI and ADAM genes might interfere with neuronal plasticity and remyelination after SCI, and further studies enhancing the upregulation of LGI and ADAM genes are needed to clarify their roles in SCI. In a previous study, GABAergic signaling through GABA receptors promoted the survival and axonal regeneration of efferent neurons after SCI in lampreys [56], and their disruption can be prevented by exercise after SCI [57]. From extensive analysis of spinal cord tissues from the very acute to chronic stages after SCI, we were able to focus on the biological process.

We found some potential drugs by network analysis using the DHUB dataset, and the list was limited to launched drugs in the clinical field. Among the potential antagonists and agonists shown in Figure 8, dextromethorphan [58], procaine [59], amiloride [60], dasatinib [61], dehydroepiandrosterone (DHEA) [62], (R)-(-)-apomorphine [63], D-serine [64], and valproic acid [65,66] have been extensively studied in clinical neuropharmacological studies aimed at modulation after neuronal damage. Dextromethorphan is mostly used as a cough suppressant and a drug for colds and coughs, but it has been reported to have neuroprotective effects against various disorders, including cerebral ischemia, epilepsy, and acute brain damage [67]. It is known that dextromethorphan attenuates NADPH oxidase-regulated glycogen synthetase kinase 3β and NF-κB activation in various infectious situations and reduces nitric oxide (NO) and ROS production to alleviate immune responses [68]. *CYBB*, *CYBA*, *NCF1*, *NCF2*, *NCF4,* and *RAC2* genes related to dextromethorphan are the NOX2 NAPDH oxidase subunit genes and major molecules in ROS production [69]. Dextromethorphan was effective in reducing neuropathic pain in posttraumatic patients [70] and a phase II clinical trial using dextromethorphan for the treatment of neuropathic pain was completed in SCI patients [71]. Procaine is a local anesthetic drug. It can attenuate neuropathic pain through JAK2/STAT3 inhibition in a sciatic nerve injury model [59], and it also blocked receptors that cause neuropathic pain by inhibiting the transient receptor potential (TRP) channel, which includes PLAU, C2, TRPV2, and C7 [59,72]. Procaine has not yet been used for neuronal regeneration in SCI patients. Amiloride is used for the treatment of edema caused by heart failure or cirrhosis and has neuroprotective effects, increased myelin oligodendrocyte glycoprotein levels and oligodendrocyte survival, and promoted functional recovery in SCI animal models [60,73]. Dasatinib for the treatment of patients with chronic myeloid leukemia and acute lymphoblastic leukemia who are positive for the Philadelphia chromosome (Ph+) has strong senolytic activity, and when combined with quercetin, locomotor function was improved in SCI animal models [61].

DHEA is one of the endogenous so-called neuroactive steroids, i.e., steroid compounds synthesized and/or active in the central nervous system, and it can be used for the treatment of thin vaginal tissue clinically. DHEA and its metabolites are known drugs that modify 5-hydroxytryptamine (5-HT; serotonin) neuronal activity through the modulation of GABA receptors [74] and promote motor recovery in spinal cord contusion models in mice [75]. Apomorphine is a potent, direct, broad-spectrum dopamine agonist that activates all dopamine receptor subtypes and can be used to treat intermittent hypomobility in Parkinson’s disease. In an in vivo study, apomorphine facilitated erectile and micturition function in SCI rats [63,76,77]. D-serine is a key signaling molecule used by neurons and astrocytes in the mammalian CNS [64], and clinical trials are being performed for the treatment of prodromal symptoms of schizophrenia [78]. Since fine-tuning of extracellular levels of D-serine is necessary for the maintenance of precise NMDA receptor function, D-serine might be a candidate for one of the potential drugs to modulate its drastic reduction in SCI [64]. Valproic acid (VPA) is an anticonvulsant drug and is also used for prophylaxis of migraine headache and treatment of manic episodes in patients with bipolar disorder [79]. VPA is a histone deacetylase inhibitor that affects the recovery of hyperacetylation and the reduction in the inflammatory response and has neuroprotective effects on histological and motor function recovery after SCI in animal models [65,66]. Nevertheless, clinical trials of VPA in SCI patients for the reduction of chronic pain in 1994 [80] and for functional recovery in 2019 [81] failed to reveal its efficacy in the clinical field.

In addition to drugs with known effects that are widely applied to neurological disorders, including SCI, other potential drugs were newly identified through network analysis (Figure 8A,C, and Table 1). Representative examples include bosutinib for the treatment of patients with Ph+ chronic myelogenous leukemia and boceprevir for the treatment of hepatitis caused by hepatitis C virus. The distinguishing feature of boceprevir from other drugs is that it mainly acts in the chronic stage, as shown in Figure 8 and Table 1. Currently, most treatment methods using animal models are effective only in the acute or subacute stages, but more than 80% of SCI patients can survive more than 12 months, and most SCI patients are in the chronic stage [82]. Therefore, drugs such as boceprevir might be a new hope for SCI patients. Nevertheless, no in vivo studies or clinical trials related to SCI have been conducted using these drugs. L-glutamic acid is one of the most common amino acids, and L-glutamate is an excitatory neurotransmitter in the CNS. The role of L-glutamic acid in the CNS is controversial. It might regulate neuronal survival and neuroplasticity, whereas it is also associated with some neurodegenerative diseases through oxidative stress, neuroinflammation, or excitotoxicity [83,84]. Because there is no study on the effect of L-glutamic acid on SCI, it is necessary to clarify the mechanisms through further research.

It is necessary to consider the genetic change according to the sex difference because we used only female rats in this study. Previous studies reported that sex differences might affect the immune response after SCI and SCI treatment responses [85]. Estrogen from female rodents might have neuroprotective effects with anti-inflammatory and antioxidant properties, but SCI reduce estradiol level and induce estrous cycle dysfunction [85]. After peripheral nerve damage, genes related to synaptic transmission were upregulated only in female rats [86], and the proliferation of microglia was prominent in male mice [87], but few studies have identified sex differences of gene expression changes after SCI with time sequence. Therefore, further study for a direct comparison between male and female is needed to confirm genetic differences which occur after SCI clearly.

Recent artificial intelligence-based drug discovery and drug repositioning techniques can dramatically shorten the preclinical research period and cost while increasing the possibility of treating various neurological diseases that were not previously available [88,89]. Several online tools have been developed and are available for free to detect repurposing drugs according to gene expression profiles [23,90,91]. Among them, the Drug Repurposing Hub, which was used for the detection of potential drugs for SCI at each time point in this study, has the largest number of compounds in neurology and psychiatry [23]. In addition, we also performed the process of verifying the drug–gene interaction network through separate software (Figure 8).

In this study, we used damaged spinal cord tissues for molecular analysis, but for the detection of supraspinal control after SCI, brain tissues after SCI at each time point are also needed. Epigenetic changes such as DNA methylation and histone modification might be another important factor for understanding regeneration and plasticity in SCI, as previously described [13]. We plan to continue efforts to elucidate molecular and biological mechanisms by conducting epigenetic analysis through ChIP seq and Methyl seq along with genetic analysis. In addition, we will also detect drug candidates for SCI treatment by confirming the in vitro and in vivo effects of the drugs discovered in this study.

## 5. Conclusions

We concluded that time sequence analysis of gene expression changes is important in the field of SCI because changes are identified in damaged spinal cord tissue according to the time sequence after SCI, and the featured biological process at each time point can explain the pathophysiological mechanism of SCI. Based on the discovery of potential antagonists and agonists, we expect to find effective treatments for SCI in the clinical field through further studies revealing the efficacy and safety of potential drugs.

## Figures and Tables

**Figure 1 cells-11-02236-f001:**
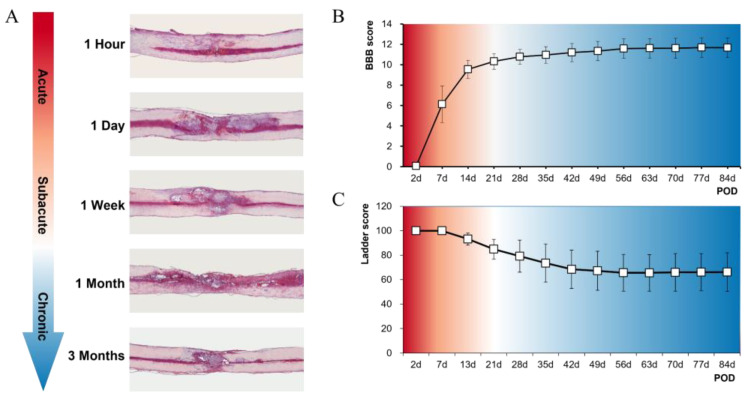
Histological and functional results of a spinal cord contusion model in rats. (**A**) representative images of hematoxylin and eosin staining at five time points from 1 hour to 3 months; (**B**,**C**) locomotor functions following SCI. Basso–Beattie–Bresnahan (BBB) scores (**B**) and ladder scores (**C**) over a 3-month period SCI model (*n* = 9) from 2 to 84 days after operation. Abbreviation: POD = postoperative days.

**Figure 2 cells-11-02236-f002:**
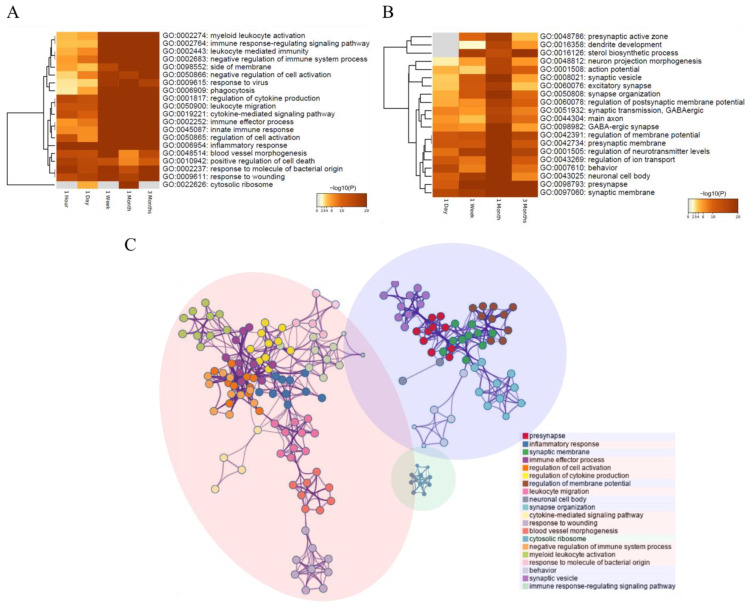
Functional analysis of DEGs after SCI treatment across five time periods using Metascape software. (**A**,**B**) The functional classification for up- (**A**) and downregulated (**B**) genes after SCI are screened and visualized. The bar chart of clustered enrichment ontology categories (GO) with a discrete color scale represents statistical significance, and white cells indicate the lack of enrichment for that term in the corresponding gene list. (**C**) A subset of representative terms from the GO cluster is converted to GO networks. The red and blue circles represent a network of GO functions involving up- and downregulated genes, respectively, and the cluster of green circles are functions that show associations on both sides. The detailed results for the functional classification and DEGs are listed in Appendix A.

**Figure 3 cells-11-02236-f003:**
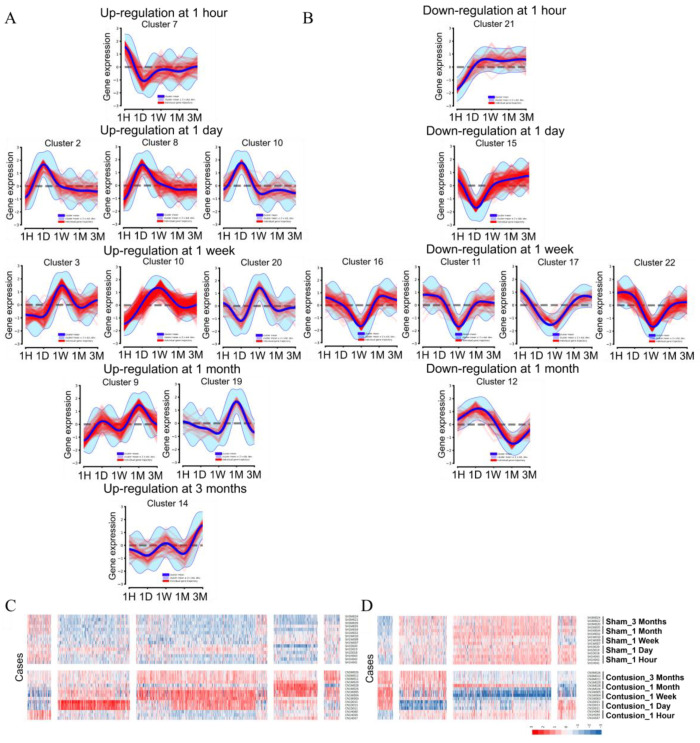
Clustered trajectories of genes in response to SCI. (**A**,**B**) Up- (**A**) and downregulation (**B**) patterns of time period-specific expression trajectories. For each cluster in (**A**,**B**), standardized log2-fold change waves (red lines) in expression at five time points are shown for individual gene trajectories, and the sky-blue boundaries indicate the posterior cluster mean ±2 standard deviations according to the cluster-specific GP. (**C**,**D**) The heatmap results with actual expression levels of each module isolated through Dirichlet process Gaussian process mixture model (DPGP) clustering of up- (**C**) and downregulated (**D**) cases of contusion models compared to sham controls. Abbreviation: 1H = 1 hour, 1D = 1 day, 1W = 1 week, 1M = 1 month, 3M = 3 months.

**Figure 4 cells-11-02236-f004:**
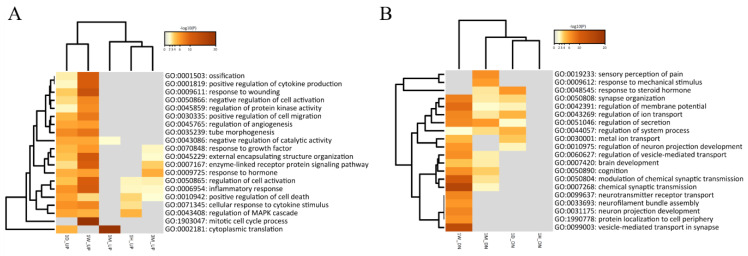
Functional classification of time-series DEG clustering by the DPGP algorithm. The functional classification for up- (**A**) and downregulated (**B**) genes in SCI are screened and visualized. The bar chart of clustered enrichment ontology categories (GO) with a discrete color scale represents statistical significance, and white cells indicate the lack of enrichment for that term in the corresponding gene list.

**Figure 5 cells-11-02236-f005:**
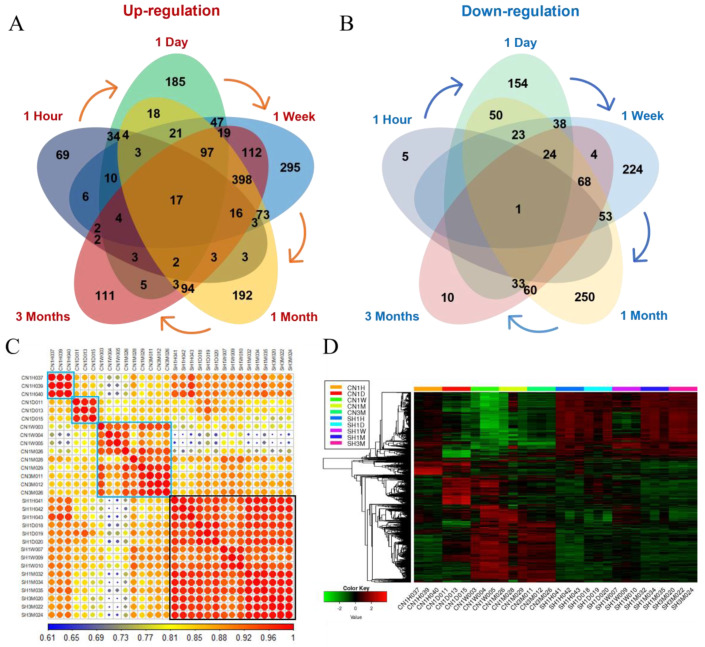
Summary of DEG analysis and significant aspects of transcriptional transition at five time points after SCI. (**A**,**B**) the number of up- (**A**) and downregulated (**B**) genes identified in the five comparison sets (contusion vs. sham cases at five time points). Overlapping areas in the Venn diagram represent genes common to every comparison group. (**C**) Pairwise Pearson correlation coefficients (PCCs) were calculated to investigate the correlation based on 2841 DEGs. A correlation matrix is gradually represented from 0.61 negative correlation (blue) to one positive correlation (red). (**D**) Hierarchical clustering heatmap for a total of 2481 DEGs identified at all time periods are represented. A histogram in the color key shows the number of expression values within each color bar. Abbreviation: CN = contusion model, SH = sham-treated model.

**Figure 6 cells-11-02236-f006:**
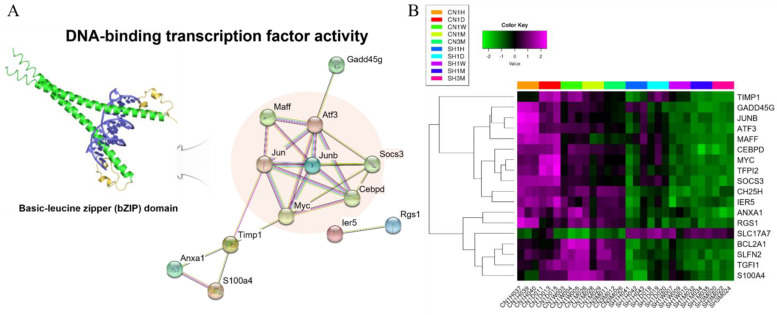
Common factors in maintaining transcriptional change over five time periods. (**A**) protein–protein interaction network based on the 18 common DEGs in all comparisons; STRING analysis mapped a network containing 13 DEGs. The remarkable genes containing basic-leucine zipper (bZIP) transcription factor domains are the seven genes in the orange circle. Each node represents a protein, and each edge represents an interaction. (**B**) The common DEGs (17 up- and 1 downregulated) showing significant changes after SCI are visualized by heatmap clustering. The legend and top color key show the experimental cases and the number of expression values, respectively.

**Figure 7 cells-11-02236-f007:**
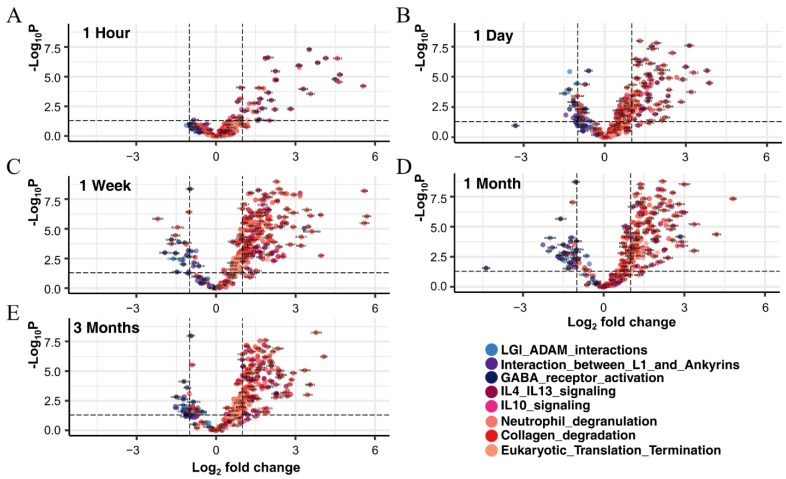
Volcano plots for gene expression according after SCI at five time points (1 hour (**A**), 1 day (**B**), 1 week (**C**), 1 month (**D**), and 3 months (**E**). Blue and red circles represent pathways to which down- and upregulated DEGs belong. The detailed patterns of each pathway are shown in Appendix A.

**Figure 8 cells-11-02236-f008:**
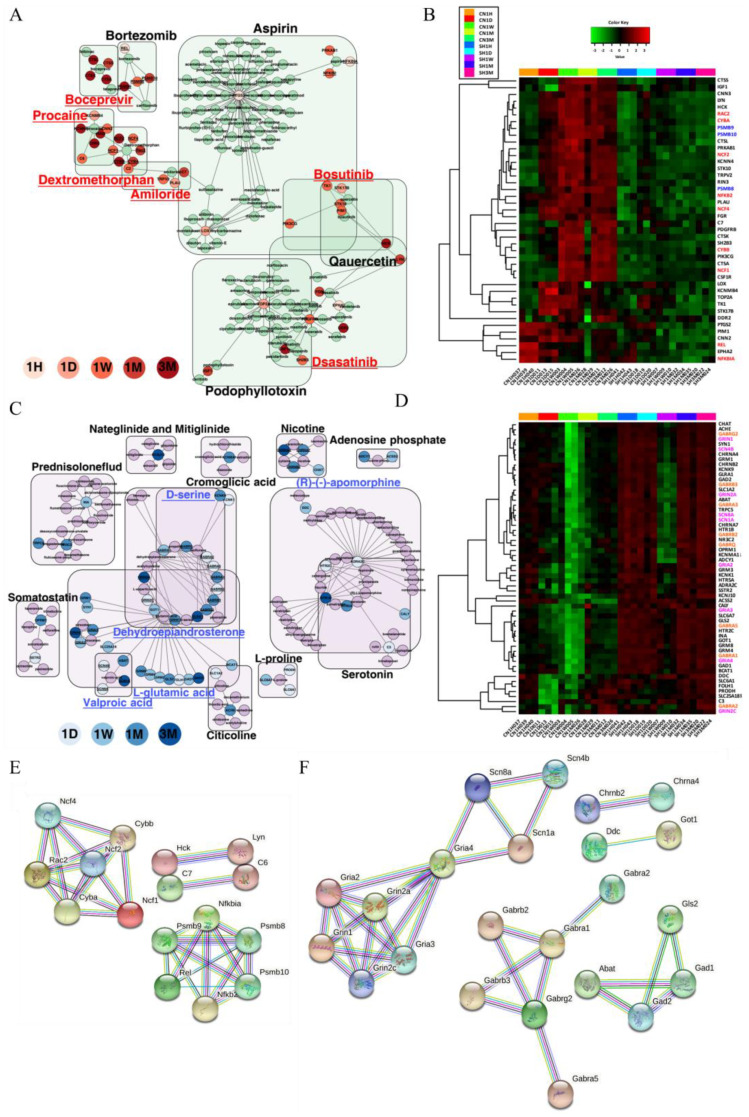
Gene–drug interaction network and PPI network. Gene–drug interaction networks involved in up- (**A**,**B**) and downregulation (**C**,**D**) and their expression patterns are shown. Up- and downregulated DEGs across the five time points are connected with the potential therapeutic drugs with specific color indications (aqua green for up and light purple for down). The up- and downregulated DEGs for each period were classified according to the red and blue gradient scales. In the expression heatmap, highly enriched genes for drug targets in the network group were color-coded. The PPI networks of all genes involved in up- and downregulated gene–drug interaction networks were investigated using the STRING online database (**E**,**F**). The detailed interactions clustered together with the gene of the drug target are listed in Appendix A.

**Table 1 cells-11-02236-t001:** List of potential therapeutic drugs interacting with the significant DEGs in SCI.

Mechanism of Action (MOA)	DEGs	Launched Drugs	Description	Genes List	Effective Period
Antagonist	Upregulated	Dextromethorphan	noncompetitive N-methyl-d-aspartate (NMDA) receptor antagonist	*C2, C6, CYBA, CYBB, NCF1, NCF2, NCF4, RAC2, RIN3*	1 d–3 m
		Procaine	HMGCR inhibitor, sodium channel blocker	*CNN2, KCNMB4, C6, KCNN4, RIN3, CNN3*	1 d–3 m
		Amiloride	sodium channel blocker	*PLAU, C2, TRPV2, C7*	1 d–1 m
		Dasatinib	Bcr-Abl kinase inhibitor, ephrin inhibitor, KIT inhibitor, PDGFR tyrosine kinase receptor inhibitor, SRC inhibitor, tyrosine kinase inhibitor	*EPHA2, HCK, LYN, PDGFRB, FGR*	1 h–3 m
		Boceprevir	HCV inhibitor	*CTSA, CTSK, CTSL, CTSS*	1 m–3 m
		Bosutinib	Abl kinase inhibitor, Bcr-Abl kinase inhibitor, SRC inhibitor	*HCK, LYN, STK10, TK1*	1 d–1w
Agonist	Downregulated	L-glutamic acid	glutamate receptor agonist	*FOLH1, GLS2, GRIN2C, GRM1, GRM3, GRM4, SLC1A2, SLC25A18, BCAT1, GAD1, GOT1, GRIA2, GRIA4, GRIN1, GRM8, SYN1, ABAT, GAD2, GRIA3, GRIN2A*	1 d–3 m
		Dehydroepiandrosterone (DHEA)	protein synthesis stimulant	*GABRA2, GABRA5, GABRB2, GRIN2C, GABRA1, GABRG2, GRIN1, GABRA3, GABRB3, GABRQ, GRIN2A*	1 d–3 m
		(R)-(-)-apomorphine	dopamine receptor agonist	*ADRA2C, HTR2C, CALY, HTR1B, HTR5A*	1 d–3 m
		D-serine	glutamate receptor agonist	*GLRA1, GRIN2C, GRIN1, GLRA1, GRIN2A*	1 d–3 m
		Valproic acid	benzodiazepine receptor agonist, HDAC inhibitor	*SCN1A, SCN4B, SCN8A, ABAT*	1 d, 1 m–3 m

## Data Availability

The data presented in this study are available from the corresponding authors upon reasonable request.

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
