# Peer review of "The Time Sequence of Gene Expression Changes after Spinal Cord Injury"

_cells, 2022, doi:10.3390/cells11142236_

Round 1
Reviewer 1 Report
The is a very interesting descriptive study that details genetic changes at different timepoints, up to 3 months, after a moderate severe contusion SCI at the thoracic level using rats. Moreover, the authors expand on a gene-drug network analysis to identify potential treatments for future clinical studies.
The manuscript is well written and the experimental design is sound and clear. Nevertheless, minor changes would improve the quality and impact of this article:
1- Explain how contusion injuries were examined to verify reproducibility between animals other than determining BBB scores at day 2 (e.g. symmetry of bruising, actual impact force plotting). Would it be possible to add videos to supplemental material.
2- Introduction should expand on previous studies on rat SCIs, and a last paragraph should be added to detail the main findings.
3- Horizontal ladder needs an expanded explanation. Are these foot-placement errors being plotted? How do you score them?
4- I would add significant value to the study if some genes within clusters could be validated by RT-qPCR.
5- Animal exclusion criteria. What are the final numbers of animals used for the study. How many were excluded per timepoint?
6- How were the three samples for RNAseq selected. Based on what parameters? was it random and blinded, or considering behavior scores?
7- On Fig 3: Please expand on criteria of cluster selection. It is not clear why 10 modules were chosen if only 5 go up while only 4 modules go down (manuscript lines 317; 318 and 321). Discuss there are no clusters with down-regulation at 3-months.
8- Please comment on possible sex differences. Which factors could give variability to studies performed in males?
Overall, this study provides new data regarding gene modulation after SCI in rats. It will also serve as a starting point for developing new treatments for SCI population. As is, the manuscript needs minor revisions to be published.
Reviewer 2 Report
Mun et al provide an extensive and detailed analysis of the transcriptome at 5 relevant points after spinal cord contusion. Gene Ontology and Reactome analyses reveal several biological categories and processes enriched among the differential expressed genes. Clusters of genes are described to change during the temporal points. Gene-drug network analysis results in a list of putative drug with possible therapeutic use after spinal cord damage. No functional studies or expression analysis of individual genes are presented. Results are clearly described and conclusions are supported by the transcriptome data. However, it would be useful to highlight what are the main novel points of the work, compared to previous transcriptome analyses in spinal cord injuries models. Immune system genes have been described to change after spinal cord injury, so I expected this result.
Specific comments:
In methods, it is not clear if libraries were prepared from a single animal or a group of animals. This is relevant to evaluate the variability among animals.
Are the pathways and biological categories enriched at these time points found in other analysis of gene expression? What are the new points found in this work?
Why authors select the DHUB dataset? Are there other similar databases?
English style should be revised. Specific points:
"Genetic changes" is normally used for changes in the genome (mutations, polimorphisms). I think authors should use "Gene expression changes" thoughout the manuscript.
Specific sentences such as "stage of chaos", "dramatic changes", "time points were implicated in the text mining", "volcanic changes" must be revised.
Reviewer 3 Report
In this manuscript by Mun et al the authors perform a transcriptomic analysis using RNA-seq, of spinal cord injury in a rat model, using contusion as the injury model. They isolated samples 1 hour, 1 day, 1 week, 1 month and 3 months after injury, covering both acute and chronic periods, as they indicate that most chronic changes occur by 3 months after injury.
The authors performed an extensive bioinformatics analysis of differentially expressed genes using the following complementary methods: 1) Gene ontology enrichment of differentially expressed genes according to their fold change when comparing contused v/s sham operated; 2) Gaussian process time-series clustering analysis followed by enrichment analysis using gene ontology, STRING and Reactome databases ; 3) Gene-drug network analysis. These analyses provided robust evidence on the time-dependent response to spinal cord injury (SCI) of the inflammatory response and synaptic transmission. They further identified novel drug candidates including dextromethorphan, procaine, amiloroide and dasatinib for future preclinical studies of the use of these drugs for the treatment of SCI. Importantly, they identified boceprevir as a candidate drug for treatment during the chronic stage.
Overall, the results presented by Mun et al are robust, the analyses performed thorough, and will be of great interest to the SCI community. I only have minor comments regarding the manuscript.
Minor comments:
Page 13, line 409. The authors wrote: “Interestingly, we proved that 12 of the 17 upregulated genes were closely interesected…”
I suggest replacing the word “proved” for “found” as it is an overstatement.
Page 14, line 434. “Figure 6. A remarkable commonality in maintaining transcriptional change over 5 time periods.”
The terms “remarkable commonality” should be replaced for more appropriate terms.
Page 14, line 465. “… and the changes in related key factors were clearly identified.”
Should include which factors they refer to.
Page 15, line 467. “Figure 7. Volcanic changes in gene expression according after SCI at…”
“Volcanic changes” should be replaced by “volcano plots” and the wording on the figure title revised.
Page 23, line 743. “In addition, we will also develop effective therapeutics for SCI by confirming the in vitro and in vivo effects…”
“Develop effective therapeutics” is an overstatement as the road from identified drugs to effective SCI therapy is very long and while drug candidates are promising, it is possible that they may not be successful in treating SCI.
